# Companion Rescue and Risk Management of Trekkers on the Everest Trek, Solo Khumbu Region, Nepal

**DOI:** 10.3390/ijerph192316288

**Published:** 2022-12-05

**Authors:** Carina Cerfontaine, Christian Apel, Daniela Bertsch, Maren Grass, Miriam Haunolder, Nina Hundt, Julia Jäger, Christian Kühn, Sonja Museol, Lisa Timmermann, Michael van der Giet, Simone van der Giet, Knut Wernitz, Volker Schöffl, Audry Morrison, Thomas Küpper

**Affiliations:** 1Department of Occupational, Social and Environmental Medicine, RWTH Aachen Technical University, 52074 Aachen, Germany; 2Department of Biohybrid and Medical Textiles, Institute of Applied Medical Engineering, Helmholtz Institute of Biomedical Engineering, RWTH Aachen University, 52074 Aachen, Germany; 3Department of Operative Dentistry, Periodontology and Preventive Dentistry, Rheinisch-Westfälische Technische Hochschule (RWTH) Aachen University Hospital, 52074 Aachen, Germany; 4Department of Internal Medicine, Ilmtalklinik, 85276 Pfaffenhofen, Germany; 5Department of Orthopedic and Trauma Surgery, Sportsorthopedics and Sportsmedicine, Klinikum Bamberg, 96049 Bamberg, Germany; 6Department of Trauma Surgery, Friedrich Alexander University Erlangen-Nuremberg, 91054 Erlangen, Germany; 7School of Applied and Clinical Sciences, Leeds Becket University, Leeds LS1 3HE, UK; 8Section of Wilderness Medicine, Department of Emergency Medicine, University of Colorado School of Medicine, Denver, CO 80045, USA; 9Royal Free London NHS Foundation Trust, London NW3 2QG, UK

**Keywords:** first aid knowledge, trip preparation, self-assessment, acute mountain sickness, risk management

## Abstract

Background: Trekking to high-altitude locations presents inherent health-related hazards, many of which can managed with specific first aid (FA) training. This study evaluates the trip preparation, FA knowledge, and FA self-assessment of trekkers (organized by tour operators vs. individually planned tours). Data obtained shall be used for specific FA trip preparation and management of emergencies en route for this population. Methods: A total of 366 trekkers on the Everest Base Camp Trek, Nepal, were interviewed using a questionnaire specifically designed to evaluate their FA knowledge and management of emergencies. Data evaluation was performed using descriptive statistics. Results: A total of 40.5% of trekkers experienced at least one medical incident during their trip, of which almost 50% were due to acute mountain sickness (AMS). There was more AMS in commercially organized groups than in individually planned ones (55% vs. 40%). For more than 50%, no medical care was available during their trip. A total of 80% could answer only 3/21 FA questions completely correctly. Only 1% showed adequate knowledge concerning FA strategies. A total of 70% were willing to enroll in an FA class specialized towards the needs of trekkers. Conclusions: The importance of high-altitude FA knowledge and trip preparation is widely underestimated. There is an unmet demand amongst trekkers for specific wilderness FA classes.

## 1. Introduction

Trekking to remote global locations has increased exponentially over the last two decades and, according to the analyses of the World Tourist Organization (UNWOTO), this trend will continue [1]. In contrast, data concerning a trekker’s trip preparation with special regard to medical emergencies en route and risk management are scarce. Amongst mountaineers climbing “classic” routes in the European Alps, it has been shown that there are significant deficiencies in their first aid (FA) knowledge, especially concerning companion rescue in an “alpine” environment, commonly referred to as the “wilderness” [2]. “FA knowledge” in the context discussed here includes all medical, tactical, and technical know-how a person should have to provide sufficient FA in the respective environment. Lechner obtained similar results when she investigated trekkers on the Annapurna Trek [3,4]. This suggests that the problem is caused by a lack of interest or awareness in FA as the topic itself is not too difficult or complex to learn; a medical lay person can be taught the relevant topics within a total of about eight hours of specific training [2]. On the other hand, it should be clear to any person who leaves for a trek in a remote region with minimal medical infrastructure that, in the case of an emergency, the ability of sufficient FA by their companions may mark the difference between survival and death. This is an important topic when potential trekkers seek medical advice prior to departure.

This study aims to analyze the data obtained from trekkers en route in order to inform and improve future FA training courses for this population. It also aims to verify the data of Lechner, as mentioned above, as well as those of Scharfenberg, because it is well-known that the country of origin of the trekkers differs significantly across certain routes [4,5]. With more French and Italian trekkers in the Annapurna region [4,5] and more Germans, British and American ones in the Everest Region [6,7], Lechner’s data may not be sufficient to establish a FA curriculum for the general trekker. There are significant differences in the awareness of FA and emergency planning between different populations [8].

This study questioned 366 trekkers at high altitude on the Everest Base Camp (EBC) Trek about their trip preparation, emergency incidents during their trip, including high altitude problems, FA knowledge, and self-assessment concerning FA skills.

## 2. Materials and Methods

This cohort study assessed trekkers on the Everest Base Camp Trek in the Solo Khumbu region in the Nepalese Himalaya. Trekkers were informed by signs and were also directly addressed in camps or lodges. Everybody who volunteered and met the inclusion criteria mentioned below was included. Only very few persons (*n* = 3 rsp. 0.8%) refused to join. Therefore, there was no selection bias.

Inclusion criteria were: aged 18+ years, and having sufficient language skills in German or English. Participants were asked to complete an eight-page questionnaire covering: general and demographic information (e.g., gender, age, medical experience), their trip preparation, their travel mode (organized by tour operator (org.) vs. individually planned tour (ind.)), their interest in participating in a class called “Companion Rescue and Risk Management in Trekking”, their FA knowledge (21 multiple-choice questions with five answers each, giving a maximum of 105 correct answers), and self-assessment of their actual FA skills. The medical topics were grouped (e.g., fractures, altitude diseases, etc.), and for each group the participants were asked to rate their knowledge using a Likert scale with the following five grades: very good, good, average, little and very little. Participants were also asked to inform about any professional medical training, in order to enable a differentiated data evaluation: “medical lay person” vs. “medical person”. The latter included physicians, medical students, paramedics, nurses, and mountain rescue staff.

The FA questions were modified for trekking and were based on a questionnaire used in previous studies assessing mountaineers and trekkers [4,5,8]. The questions reflected typical basic FA situations encountered while trekking. All questionnaires were anonymized, and participants were not allowed to work collaboratively.

In order to evaluate the data as correctly as possible, acute mountain sickness (AMS) had to be defined, because all symptoms of AMS may also be caused by other disease. For this study, AMS was assumed when there was a headache plus any other typical symptom of AMS and i. other reasons were excluded (e.g., sunstroke as cause of headache), and ii. the severeness of symptoms was higher than 3 points on the Lake Louise Score [9,10].

A “medical incident” was defined as any disease or injury where the patient took any drug or any other content of the medical kit (e.g., wound dressing), needed advice by medical personnel or care by local institutions (e.g., the clinic of the Himalaya Rescue Association (HRA) at Pheriche).

Data was evaluated using IBM SPSS Statistics 20^®^ (IBM, Chicago, IL, USA). Group differences were evaluated using non-parametric Mann-Whitney-U-Test and Chi-Square-Test. Significance was defined as *p* < 0.05, while 0.05 < *p* < 0.1 was carefully interpreted as tendency.

To measure the accuracy of the self-assessment compared to the actual knowledge proven by the questionnaire, both results were converted into a percentage and subtracted. This created a range from −100% to +100%, with 0% indicating a perfect match between self-assessment and FA knowledge of the respective topic. For example, −100% indicated an extreme overestimation or a complete lack of knowledge, and +100% signified an extreme underestimation of their FA skills.

The study was performed in accordance with the declaration of Helsinki and was advised by the ethical committee of RWTH Aachen Technical University, Aachen/Germany (study no. EK196/11).

## 3. Results

Of the 366 trekkers who were included in the study, 58.7% were male, and 40.4% were female (0.9% missing data). Participants were from 36 countries (*n* = 366). A total of 70.2% (*n* = 257) were on a commercially organized group trekking trip, whereas 28.4% (*n* = 104) had organized their trip individually (1.4% (*n* = 5) missing data). The average age was 38.9 years (range: 18–74 years).

Individual trekkers (mean: 36.6 years) were slightly younger than the organized trekkers (mean: 39.8 years; *p* = 0.048, Figure 1). A total of 44.2% had no trekking experience, 21.4% provided up to 5 years of trekking history, and 39% had never been above 3000 m before (no significant difference between ind. and org.). Individual trekkers were more likely to travel in small groups without a leader (*n* = 100; 83.0% ≤ 5 team members; mean: 3.7 people; 70.3% no leader), in contrast to commercial trekking groups which consist of up to 19 clients (*n* = 203; 36.5% ≤ 5 clients and 32.5% 11–15 clients; mean: 7.9 people; *p* < 0.0001) with 2–5 tour guides per group (62.6%; *n* = 203; mean: 2.3 guides; *p* < 0.0001).

Of the trekkers surveyed, 80.1% had no medical experience, 9.1% were medical students or doctors, and 10.8% worked as medical staff, meaning that 19.9% had some kind of professional medical training. A total of 56.6% had joined an FA class once, and 69.4% indicated a willingness to enroll in an FA class specialized towards the needs of trekkers. The majority would prefer a weekend class with a maximum duration of 12 h, teaching both practical and theoretical knowledge, and would be willing to pay up $50 US.

Only 58.7% stated that they had access to FA at all times during the trek, and 38.0% said they could see a doctor whenever needed. However, when asked for more detailed information on this question, 53.0% (*n* = 234) stated that there was no medical care available during their trip. Only 5.1% (*n* = 234) had a doctor in their group, 13.2% (*n* = 234) carried a FA kit in the group, and 14.1% (*n* = 234) relied on the health post along the EBC trek in case of an emergency.

A total of 56.6% had been encouraged to use medicine to assist acclimatization. Significantly more individual trekkers were advised to take these drugs compared to the commercially organized trekkers (ind.: 71.3%, *n* = 101 vs. org.: 54.0%, *n* = 250; *p* = 0.002). Out of the trekkers who provided detailed information on which medicine they were encouraged to take, 88.7% (*n* = 159) said that it was acetazolamide.

When asked about a medical emergency, illness or other accidents, 40.5% noted that there had been at least one medical incident during their trip. This occurred significantly more often in the individual trekking group (55.9%, *n* = 102; out of which 11.9% needed to see a doctor) than in commercially organized trekking (34.1%, *n* = 249; *p* < 0.0002). Most incidents were due to acute mountain sickness (AMS) (48.4%, *n* = 126), gastrointestinal problems (32.5%, *n* = 126), headache (20.6%, *n* = 126) or a cold (19.8%, *n* = 126). AMS symptoms were more likely to be found in commercially organized trekkers (54.8%, *n* = 73) than in individual trekkers (39.6%, *n* = 53; *p* = 0.093).

Individual travelers sought information on general health prophylaxis significantly more often prior to the trip than the commercially organized travelers (ind.: 81.8%, *n* = 99 vs. org.: 59.2%, *n* = 250; *p* < 0.0001). The former made plans for an early return to their country in case of an emergency more frequently than those on the commercially organized tours (ind.: 63.4%, *n* = 101 vs. org.: 36.0%, *n* = 250; *p* < 0.0001). In general, 50.5% of the participants did not take any precautions for an emergency repatriation. 19.9% relied on their insurance, and 4.4% had some spare money to finance the possible rescue and emergency return to their home. When assessing trekkers for those with travel health insurance coverage, the situation was reversed. Those on a commercially organized tour had this insurance more often than the individual trekkers (org.: 85.2%, *n* = 250 vs. ind.: 68.3%, *n* = 101; *p* = 0.0004). Of those questioned, 78.3% claimed that a timetable change of the planned tour was possible at all times, and 88.0% said that it was possible to interrupt the tour at any time to go back home.

On average, the participants scored 61.2 out of 105 points for the FA questions. A total of 17.8% were not able to answer any question completely correctly, meaning that all five multiple choice answers were marked as incorrect. The majority (80.1%) completed up to 3 out of 21 questions correctly. The highest scored questionnaire had 16 questions answered correctly (Figure 2).

When considering all the answer options individually, 53.4% (*n* = 339) scored 64 to 84 points, which equated to “good” FA knowledge. Only 1.5% had “very good” knowledge (85 to 105 points). In the subgroup analysis assessing medical experience, non-medically trained persons scored 48.5% “good”, and 40.4% had “average” results (43 to 63 points) in FA knowledge (*n* = 260). Among the medical staff, 70.6% showed “good” and 11.8% had “average” FA skills (*n* = 34), with doctors and medical students demonstrating the best results with 77.4% “good” and 12.9% “very good” FA knowledge (*n* = 31). The Chi-Square-Test confirms that people with medical experience (mean of 65.8 points) knew more about FA than those without medical training (mean of 59.9 points; *p* = 0.02). The following FA topics were answered correctly most often: head injury (39%), heart attack (36%), high altitude cerebral edema (HAPE), and frost bite (26% each). In contrast, the following topics were only occasionally answered correctly: spinal injuries (2%), FA strategies (1%) and tooth problems (1%). A total of 63.3% of those questioned were able to self-assess their knowledge quite accurately. The trekkers tended to slightly underestimate their FA skills (Figure 3).

## 4. Discussion

This study’s cohort was part of the trekking community in the Sagarmatha National Park, Nepal. According to the data supplied by the Nepalese Ministry of Culture, Tourism and Civil Aviation, a total of 57,000 trekkers visited this area with an average stay lasting 13 days [11]. Compared with other studies, our cohort also represented a typical distribution of people with medical experience (20%) and non-medically trained persons (80%) [4,5,12]. Age and gender distribution were also similar to other studies. The topic “FA” is highly relevant for organized trekkers as well as for individual ones since medical incidents happen quite often—fortunately most of them minor ones [4,5,13].

A deficiency in FA knowledge is a well-known problem among the general population and among persons travelling to remote regions [14]. The consequences of such deficiencies in mountaineers who receive only some training (physicians, guides, rescue staff) are even more serious [5,8,14]. For years now, up to 50% of all emergency calls are transmitted by mobile phone in some regions of the European Alps [15]. However, even in regions with a perfect infrastructure like Zermatt (Switzerland), it still requires a minimum of 30 min to get rescue teams on scene [16], and during this period the companions must care for the patient. This time increases to several hours or even days for trekkers or mountaineering expeditions. There is a general consensus about the benefit of a short interval between the accident and professional attendance in patient outcomes [17], and therefore, advanced FA skills of mountaineers, especially of trekkers and expedition climbers, are advantageous. To give an example, in an urban environment, 20% die from causes that could have been treated with first aid [18]. This will be even more pronounced in the cold, wet, windy, and hypobaric mountainous environment of a trekking region.

However, FA knowledge will decrease with the passage of time, and refresher trainings should be recommended. Studies show that most people fail to sufficiently recall FA knowledge one to two years after a FA class, and that the practical skills vanish more quickly than theoretical ones [19,20]. Even paramedics are not able to perform effective resuscitation one year after their last training [21]. People tend to rely on others when it comes to performing FA. A personal feeling of competence is crucial for the initiation of FA [22]. The underestimation of personal ability might lead to a complete lack of application of FA intervention.

The data presented here demonstrate that the majority of the interviewees principally had “average” or “quite good” FA knowledge with a “fairly good” self-assessment, but they generally tended to underestimate their knowledge slightly. It also shows that their knowledge was often ineffective in the wilderness, since typical FA classes are designed for an urban setting and fail to train the specific skills and strategies needed to be effective in the wilderness [23]. This is similar for medical students, as there are only a few universities worldwide which offer wilderness medicine training for students [24].

The two best answered topics were head injuries and heart attack. While it might be helpful to know how to treat head injuries in the field, knowing how to identify or treat a severe heart attack may be not. If a person suffers from a heart attack in the Solo Khumbu—independent of whether it later turns out to be a “simple” angina pectoris or a myocardial infarction without cardiac arrest or fibrillation—this person nowadays has a quite good chance of survival, since helicopter service is fairly rapidly available if the weather conditions allow such operations, and coronary intervention has been established in Kathmandu. On the other hand, if someone suffers from a cardiac arrest in the mountains, without any infrastructure, this person most likely will not survive even with the best FA care. Even in an urban setting, the survival rate of resuscitated patients is low, especially when resuscitation is necessary in trauma patients (0.5% (*n* = 562) [25]. Indeed, CPR is a very rare situation in the mountains: among 2730 rescue operations in Switzerland (Wallis) and Austria (Tirol), there were only seven resuscitations (0.26% of all operations) [8]. It may be concluded that resuscitation skills and the respective training are of less importance in the mountains.

Nevertheless, the participants had only rudimental knowledge (3–5% correct answers) in the following topics: rescue strategies, hypothermia, water disinfection, basic life support or snow blindness, all of which are essential to FA in the wilderness. Comparative studies emphasize the same lack of knowledge among trekkers and climbers [5,8,26]. Rescue strategies are of special interest in an area without infrastructure and an ambulance system. It can take up to 24 h before professional support and rescue appears due to location, weather conditions, and/or payment regulation. In some cases, for example, during the bad weather phase which blocked thousands of people in Lukla in 2011, it may take several days or more than a week to get a patient out. Independent from other diagnoses, it may be vital to keep the patient warm for this period. Némethy et al. stated that frost bite is the most common problem at the Himalaya Rescue Association (HRA) health post at Everest Base Camp [27]; and, according to Küpper, all victims in the Western Alps were hypothermic when the rescue helicopter reached them [2], even though the Western Alps have good rescue infrastructure. Therefore, detailed knowledge of hypothermia and frostbite are essential FA knowledge for trekkers.

Only 18% of the interviewees knew that a slow ascent effectively prevents acute mountain sickness. This is highly alarming given the fact that nearly half of this collective suffered from AMS symptoms and that a slow ascent is one of the easiest and most effective ways to prevent AMS [28,29,30]. This lack of knowledge seems to be a common problem in mountaineering and climbing worldwide [5,31]. The UIAA recommends sleeping no higher than 300 to 500 m higher than the previous night, starting at 2500 m, and planning an extra rest day every 1000 m of altitude gain in order to give your body enough time to acclimatize to the increasing altitude [32], and before booking it should be checked whether the tour operator follows such recommendations [33]. Four out of five of the most frequently named trekking organizations in this study did not follow these recommendations in their trip schedule, even though it would be easily possible—as one organization demonstrated. Although our data do not provide the exact number of persons who were put at risk of altitude disease by inadequate altitude profiles, it can be stated that this represents a significant portion of all trekkers visiting the Solo Khumbu area. This supports Eggert´s findings that 50% of the questioned organizations did not think that such a safety strategy was feasible [34]. Of the questioned organizations, 80% claimed to provide sufficient information on AMS to their clients [34]. This matches the current data where almost three quarters of the interviewees were willing to gather information on AMS before traveling to this region. Nevertheless, basic knowledge was not able to be recalled when asked. There were only 14.9% correct answers on acute mountain sickness in this study. In contrast, Bauer showed that about 60% of travelers in Peru could remember parts of what they were taught about general health prophylaxis right before traveling [35].

The difference in AMS incidence between organized groups and individual trekkers may be a consequence of the more fixed schedule of organized groups. This was described years ago [36], however, things did not change significantly despite many recommendations being easily available (e.g., [37]). Although our data cannot prove the higher incidence in organized groups, it is obvious that these groups take acetazolamide or other drugs to prevent AMS much more frequently than individual trekkers.

A topic rarely—may be never—addressed in FA trainings is acute dental problems. Küpper et al. published data from the Annapurna Trail showing that the occurrence of dental incidences is 1:23.7 person days on trek, with severe problems encountered as follows: 1:145.2 person days of dental pain, 1:339 person days of lost fillings, and 1:509 person days of dental fractures [38]. Taking into account that trekking is a group activity in most cases, with typically 3–4 members in treks organized by individuals, and 8–10 in organized travelling groups, it is very likely that dental problems will occur in such groups while en route. Therefore, FA for dental problems is essential for trekkers and should include at least basic knowledge on how to deal with dental pain, lost teeth, broken teeth, caries, or lost fillings [38,39]. Such topics are not included in normal (“urban”) FA training and consequently the trekkers’ knowledge in the actual study was virtually zero.

Kühn pointed out that a recently completed FA class improved knowledge in almost all categories [26]. These findings emphasize the importance of an FA class specializing in the needs of trekkers in the wilderness and at high altitude. Kühn also designed a schedule for specialized first aid training for multiple disciplines of outdoor and climbing sports [26]. With such a modular training, each climber may choose the modules which are of special interest for his individual activities, e.g., sports climbers are not interested in altitude diseases or cold injuries, while mountaineers or trekkers are. The actual study supported her hypothesis concerning the need to emphasize specific topics. For trekkers and expedition climbers these include basics in prophylaxis and treatment of altitude diseases, environment-related problems such as drinking water hygiene, snow blindness, lightning, frost bite, hypothermia, basic rescue strategies depending on the terrain, treating teeth problems as well as soft tissue and bone injuries, and last, but not least, correct handling of the first aid kit and its medication.

For decades now, a well-known problem has been how to address the target group and how to motivate these people to join specialized FA trainings [40,41,42,43]. Interestingly, those with higher individual risk do not tend to be interested in first aid training [2,8,42]. This could also be true for trekkers. However, observational data from trainings for the German Alpine Club Aachen (Küpper T., unpublished data) shows that such psychological barriers are much lower if the target group has the feeling that the training was specifically designed for their activity. A modular system for the seminars, whereby special topics are offered in addition to a basic training for all, offers the best flexibility to specifically address mountaineers of different disciplines [26].

Any course also includes plans for specific training methods. There are different opinions concerning the didactic strategy needed for FA courses for the wilderness. First of all, students need to be taught awareness of a known problem, and they must be offered a solution, and an explanation of the solution [44,45]. Time limitations force a choice between stressing the principles and practicing first aid. Both Taubenhaus and Rettig favor teaching principles, because understanding will produce more confidence compared to an emphasis on training techniques without understanding [46,47]. Courses should include scenario-based training whenever possible [48,49].

This study’s findings support those of Küpper et al., suggesting that the total time of the FA training should not exceed 1.5 days, e.g., a weekend, or 3–4 evening sessions [8]. Otherwise, the acceptance of such a course and therefore the number of trained trekkers will be limited [8,26,42]. No data is available as to when a refresher course for trekkers or alpinists in general should be recommended. A significant decrease in resuscitation skills occurs after 1 year [50,51,52,53]. In paramedics the loss was only a little less [21]. Based on these studies, and taking into account the problems of motivating people to join a course, we would recommend a refresher course for mountaineers every 3 years [8,26,53].

## 5. Limitations of the Study

As with any open-field study, it should be proven whether the collective represents the target group of trekkers in total and whether the collective is similar to those of other studies. Age, gender, experience in mountaineering, and professional medical training fits well with the collective of several other studies (e.g., [4,5,8] and the studies referenced in these papers). Differences are caused by nationalities. However, taking into account that there is no difference in the main outcomes between earlier studies, especially between Lechner’s and Scharfenberg’s at the Annapurna Circuit [4,5] and the actual one, this should be negligible.

Since all participants joined the study independently of whether or not they had experienced a medical problem during their trip, any recall bias about incidences should be minimal. This study focuses on FA and is mainly aimed at medical lay persons. Such FA does not include rare diseases and their treatment. Therefore, a collective of 366 participants should be sufficient.

## 6. Further Work to Be Done

To improve the situation and to make trekking safer, several tasks should be followed up in future. Firstly, trekking organizations should perform a careful review of the altitude profiles of their tours. Many profiles include guests sleeping at altitudes of 3500 to about 4000 m on the second night (Küpper T, unpublished data). This is far too fast too high, and really provokes altitude disease [32]. According to the Himalayan Rescue Organization and to [4,5,6], such diseases make up to 65% of all medical problems in the Himalayas, and up to about 6000 m they can be avoided nearly completely by proper acclimatization [32]. Doctors advising people who are planning a trek should also always carefully check the respective altitude profile.

Since proper acclimatization needs some days, there is a common argument by the travel industry that this will increase costs and people will not be willing to pay a higher price for the travel. However, since we are talking about well-established strategies to keep clients healthy, any violation against these rules may be called “bodily injury caused by negligence” or even “aggravated battery”, thus policy makers should prioritize adequate planning of tourism and travel product quality.

Any person or institution who is interested in safety in the field, e.g., the alpine clubs, should establish courses to train hikers or mountaineers in field-related FA. To best address the different target groups this should be organized in a modular system [26]. This may also include downloadable versions or online training for theoretical topics. Of course, such trainings are unable to replace practical trainings. Mass media may support the readiness of the target groups to join FA trainings [54]. Electronic information systems are also more and more valuable in assisting FA providers on scene. However, it must be possible to pre-download such software due to the lack of internet access in several mountain regions. And—last but not least—it should be noted that sufficient FA in the mountains does not just include medical knowledge. Trekkers and mountaineers should be competent in at least basic companion rescue techniques, since sometimes emergencies happen in places that are not easily accessible or where it is difficult to perform FA. Such techniques include also self-rescue, e.g., from crevasses.

Since it is well established nowadays what happens in the field and what to do in the case that FA is necessary, scientists should focus on topics such as didactical strategies for addressing target groups, or which is the best interval to recommend a refresher training.

## 7. Conclusions

FA knowledge is limited amongst the growing trekking community and does not provide the additional specialized knowledge essential for dealing with medical emergencies in remote regions. To make this sport safer, trekkers and trekking companies/organizations should target and inform this population that specific FA training is necessary. Such training should be updated regularly, at least well in advance of the next tour. The following topics should be emphasized in the FA training: rescue strategies, hypothermia, prevention of disease by water disinfection, basic life support, snow blindness, dental problems, and altitude diseases (especially AMS), acclimatization and altitude profiles.

## Figures and Tables

**Figure 1 ijerph-19-16288-f001:**
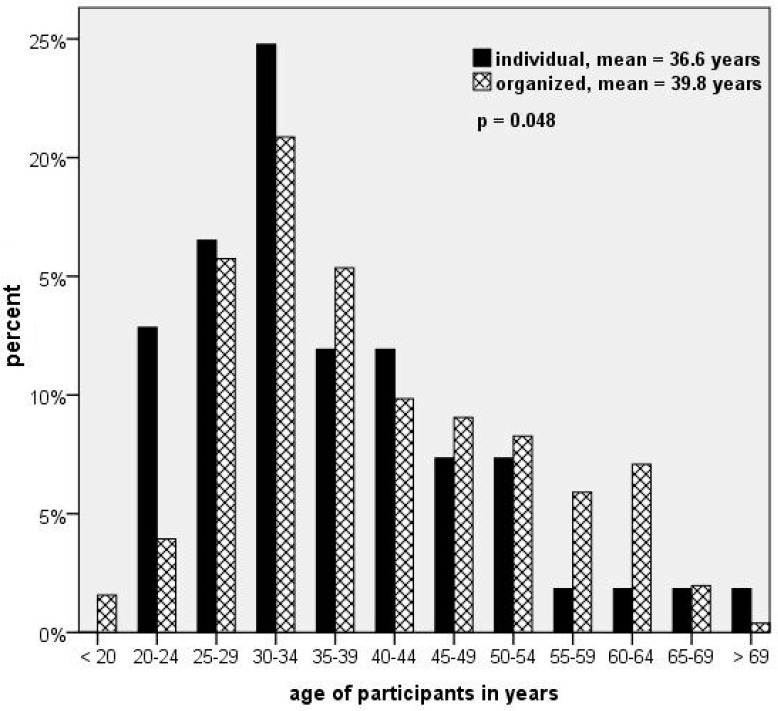
Age of the participants and mode of travel.

**Figure 2 ijerph-19-16288-f002:**
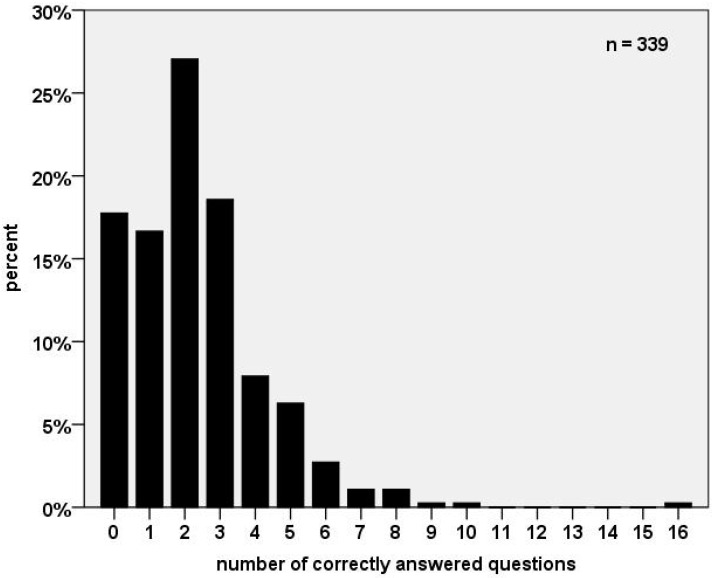
Number of correctly answered first aid questions.

**Figure 3 ijerph-19-16288-f003:**
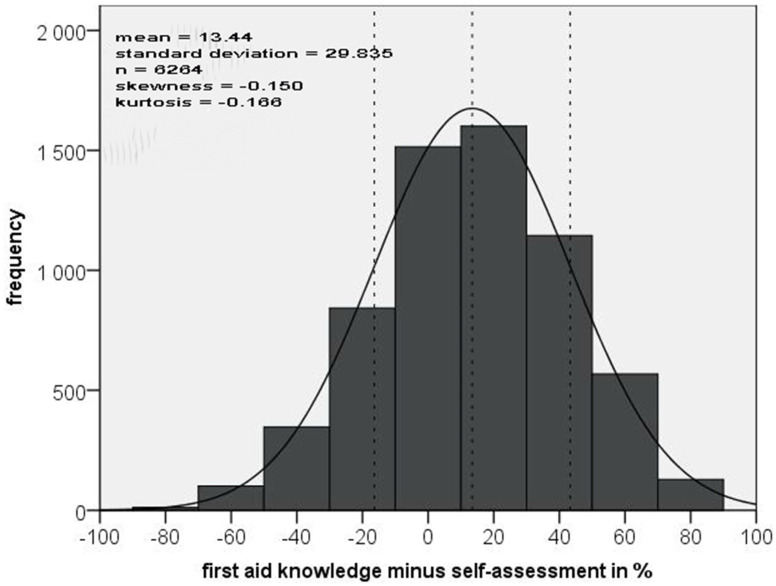
Number of correctly answered first aid questions vs. self-assessment.

## Data Availability

Data are available from the authors.

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
