# Peer review of "Companion Rescue and Risk Management of Trekkers on the Everest Trek, Solo Khumbu Region, Nepal"

_ijerph, 2022, doi:10.3390/ijerph192316288_

Round 1

Reviewer 1 Report

Thank you for the opportunity to review the paper titled: Companion Rescue and Risk Management of Trekkers on the Everest Trek,

Solo-Khumbu Region, Nepal. The purpose of this paper is to propose and evaluate trip preparation, FA knowledge, and FA self-assessment of trekkers. The topic area needs to be studied more, likely due to the need for a well-delineated conceptualization of digitalization that contributes to trip preparation performance through supply chain integration with external parties.

Thus, this study is timely. The paper presents a current and relevant theme. The theoretical framework is consistent, well used, and provides an appropriate basis for the construction of the methodology. The results obtained are relevant and present a contribution to their field of study. Overall, this is an interesting study. I have provided detailed feedback in the attached file.

Introduction

Could you define keywords such as” first aid knowledge”?

The background of the study needs to sufficiently discuss the current status of the problem.

Therefore, it is not possible to grasp the importance of the study and the research gaps.

It is recommended that the researcher should be specific about the importance of the issue.

Results

The results were clearly presented and properly analyzed. The conclusions adequately brought together all the other elements of the study.

Discussion

I would like to see greater explanation of what the findings mean for future research, practitioners (i.e. tourist industry), and policy makers.

  The current suggestions feel too vague. I would like to see the discussion of the findings to include more on answering the ‘so what?’ question.

Author Response

see attachment, please!

Reviewer 2 Report

This study examined the first aid training of trekkers on the Everest Trek. The information reported was gathered by surveying 366 trekkers. The paper is well-written, and the topic is important. However, I have a few comments.

1)     The authors should consider including the questionnaire as a supplement.

2)     The percentage of individual and commercial trekkers is presented. The count would be helpful too. (first paragraph of the results)

3)     The number of different commercial companies should be provided. For example, line 269-270 mentions “four out of five.” Were there only five commercial trekking companies?

4)     Often multiple references are shown as [X] [X] [X]. Instead, these should be [X-X]. Please correct this.

5)     Line 187. I believe the word cohort is singular, and the word cohorts is plural. The verb tense should be corrected.

6)     Line 269-270. The support of Eggert’s findings that most commercial organizations “did not think a safety strategy was feasible” is too strong. What data in the current study is presented to support this? There may be other reasons the trekking organization did not follow this recommendation.

7)     Line 273-275. Is the 80% referring to the data in this study?

8)     Lines 288-299 discussed dental problems. Did the survey include questions on this topic? I don’t see any mention of this in the results.

9)     Line 343-345. This sentence is cumbersome, and I am not sure of the message.

10)  Line 351-353. I am not sure the point of this sentence.

11)  Line 361. Emphasized is spelled wrong.

Author Response

see attachment, please!

Reviewer 3 Report

A study on knowledge and skills in the field of providing assistance, especially in extreme situations that occur in high-mountain tourism, is extremely valuable. The results should be widely disseminated to interested individual and commercial trekking groups to understand the severity and importance of the problem.

The work has the correct content layout, but I would suggest that the Discussion should not include any issues that were not discussed in the Introduction and Results. In the Discussion, the authors are to verify the hitherto achievements of science presented in the Introduction and relate them to their research. Therefore, I propose to include some of the content from the Discussion based on the publications of other authors of the first aid problem in the Introduction, and to confront its most important aspects with your results in the Discussion.

I also propose to clearly articulate the purpose of the research and research questions in the Introductory Part.

At the end of the Discussion, it would also be beneficial to propose to trekking organizations and individual mountain tourists a strategy for acquiring knowledge and skills to provide specialist assistance before going to high mountains.

The proposition of how to use the knowledge about self-rescue and helping others, e.g. on electronic devices adapted to high mountain conditions, would also be significant.

Author Response

see attachment, please!

Round 2

Reviewer 1 Report

The authors have been able to communicate in the text why it is necessary to publish this manuscript. Thus, the paper contains appropriate new and significant information that warrants it's publication in the IJERPH.